# Insecticidal Activity of Selected Essential Oils against *Drosophila suzukii* (Diptera: Drosophilidae)

**DOI:** 10.3390/plants12213727

**Published:** 2023-10-30

**Authors:** Dragana Bošković, Slavica Vuković, Sanja Lazić, Nuray Baser, Dušan Čulum, Dragana Tekić, Antonije Žunić, Aleksandra Šušnjar, Dragana Šunjka

**Affiliations:** 1Faculty of Agriculture, University of Novi Sad, Trg Dositeja Obradovića 8, 21000 Novi Sad, Serbia; slavica.vukovic@polj.edu.rs (S.V.); sanja.lazic@polj.uns.ac.rs (S.L.); dragana.tekic@polj.uns.ac.rs (D.T.); antonije.zunic@polj.edu.rs (A.Ž.); aleksandra.susnjar@polj.edu.rs (A.Š.); 2CIHEAM-IAMB—International Centre for Advanced Mediterranean Agronomic Studies, 70010 Bari, Italy; baser@iamb.it; 3Faculty of Science, University of Sarajevo, Zmaja od Bosne 33-35, 71000 Sarajevo, Bosnia and Herzegovina; dusan.culum@pmf.unsa.ba

**Keywords:** *D. suzukii*, essential oil, oviposition deterrence, bioinsecticidal effect, repellency

## Abstract

The spotted wing drosophila (SWD), *Drosophila suzukii* Matsumura (Diptera: Drosophilidae), is an invasive polyphagous pest of soft-skinned fruit that has started to threaten small fruit production in Europe since 2008. High reproductive capacity, short generation time, and difficulties in visualizing early damage contribute to its rapid spread. Currently, the control strategy against *D. suzukii* mostly relies on treatment with synthetic insecticides. Keeping in mind that this pest causes the greatest damage during the harvesting period, control using chemicals is not recommended due to the increased risk of high pesticide residue levels in the fruit. With the aim of reducing the use of insecticides, there is a need for developing an environmentally safer way of control. Alternative solutions could rely on the use of essential oils (EOs), which can be used in conventional and organic production systems. Four essential oils, geranium (*Pelargonium graveolens)*, dill (*Anethum graveolens*), Scots pine (*Pinus sylvestris)*, and bergamot (*Citrus bergamia*), were assessed for their insecticidal effects using four different tests (contact toxicity, fumigant-contact toxicity, repellent effect, and multiple-choice tests). The EOs applied were dissolved in acetone at three different concentrations. The most promising one was the geranium essential oil, which had the best effect in all conducted tests, even at the lowest applied concentration. Geranium oil caused very high mortality (95%), even at the lowest concentration applied, after 24 h in the fumigant-contact test, and it caused a mortality of over 85% in the contact test. It was also noticed that geranium demonstrated a deterrent effect by repelling females from laying eggs for four days after at the lowest applied concentration. Scots pine and dill EOs have moderate to strong effects in most tests. The mortality of 100% was achieved for the highest applied concentration in the fumigant-contact and contact tests. Bergamot EO did not have any significant insecticidal activity. Geranium, Scots pine, and dill have great potential to be used as an environmentally friendly way of controlling *D. suzukii* as they exhibit deterrent, repellent, and insecticidal effects.

## 1. Introduction

*Drosophila suzukii* (Matsumura), also known as spotted wing drosophila (SWD), is an invasive and harmful fruit pest native to East Asia that has spread into many European countries [1,2,3]. The main way of introduction was through the global trade of fruit infested by eggs or larvae [4]. It was first recorded in 2008 in Europe in Spain and Italy [1,2,5]. Swiftly after that, it has spread to other parts of the European continent [1]. In the territory of the Republic of Serbia, the presence of SWD was recorded for the first time in 2014 [6].

SWD is a polyphagous pest that damages soft-skinned fruit with a broad range of stone and berry crops like raspberries, blueberries, strawberries, blackberries, cherries, and grapes [7,8,9,10]. This pest can also attack wild non-crop plants from different genera (*Prunus*, *Rubus*, *Sambucus*, and *Vaccinium*) [7]. In contrast to other similar species of Drosophila flies that prefer rotten, overripe, or decaying fruits, this pest oviposits in ripening fruit, which causes significant economic loss in the fruit industry [4,11]. SWD has a unique serrated ovipositor that allows them to pierce the skin of undamaged and ripening fruit in order to lay eggs [12,13]. The symptom of the attack can be determined by examining the fruits, in which soft spots and collapse of the berry structure can be observed. Such fruits become wrinkled, change color, and have higher sugar levels, which is accompanied by softening of the skin [7,14,15]. Direct damage to fruit is accomplished through the process of oviposition as well as larval development and feeding on the fruit pulp, which leads to the decay and collapse of fruits before harvesting [7,16,17,18]. Infested fruit is susceptible to secondary infections by other pests and pathogens like bacteria, fungi, and yeasts, which affects the faster decaying of fruit [7,19,20,21,22]. Depending on the climatic conditions, this pest develops up to 13 generations per year, which contributes to their rapid spread [7,12].

With the aim of reducing the SWD population in the field, the current strategies for controlling SWD rely heavily on the application of synthetic insecticides [23]. Considering the numerous generations and short generation time during the season, to maintain a low level of pests, it is necessary to apply synthetic insecticides multiple times during the ripening period to prevent fruit loss [4,5,24,25]. However, frequent use of synthetic insecticides leads to resistance development in the SWD population [26,27] and the occurrence of chemical residues in fruits [28]. In addition to this, these compounds can pose negative and adverse effects on the environment, such as on beneficial organisms [29,30,31]. Considering all of the above, there is an intense need for new alternative control management tools against SWD that can be implemented in integrated and organic crop production [15,25,32,33]. The current focus is aimed at natural product-based strategies, like the use of botanical insecticides that contain essential oils (EOs). These compounds are biodegradable and ecologically safer, with low or zero effect on beneficial organisms, reduced impact on the environment, and little or no residues left in fruit [34,35,36,37]. EOs have multiple modes of action compared with synthetic insecticides, which contributes to them preventing the emergence of resistant insect populations [38,39,40]. These compounds are a complex mix of volatile chemicals that can act like fumigants and contact insecticides [41,42]. They can also impact pest behavior as an oviposition deterrent, repellent, or even as attractants [43].

This study aimed to evaluate the insecticidal and behavioral effects of four essential oils on *D. suzukii*. Using different tests (fumigant-contact toxicity, contact toxicity, repellent effect, and multiple-choice), the effects of EOs were evaluated on male and female adult flies.

## 2. Results

### 2.1. Mortality of Adult Flies in Fumigant-Contact Toxicity Bioassay

In the combined fumigant-contact test, the flies were in contact with the treated surface and volatiles from the oil. The oil with the highest efficacy was geranium EO, with a mortality rate of 95%, even at the lowest concentration (1%), after 24 h. On the contrary, the bergamot essential oil did not show an effect on the mortality of the adults in all three applied concentrations. Dill and Scots pine essential oils exhibited mortality rates of over 92% at the concentration of 5%; however, in the lowest applied concentration (1%), they were below 40% (Table 1). There was a statistically significant difference between both controls and all three applied concentrations (*p* < 0.05) for geranium and Scots pine EOs. A significant difference was also observed between the concentrations of 1% and 5% and between the concentrations of 1% and 10% for the Scots pine and dill EOs (*p* < 0.05). For dill oil, a significant difference was observed between both controls and the concentration of 5% and 10%. No significant difference was observed for bergamot oil. Female flies were slightly more resistant to treatments compared with males, but not on a statistically important level (Table 2).

### 2.2. Mortality of Adult Flies in the Contact Toxicity Bioassay

Similar results were obtained for the contact test as well (Table 1). The treatment with geranium essential oil achieved a mortality rate of over 85% for the lowest concentration applied (1%) and over 92% for the other two applied concentrations. In the same concentration (1%) of dill EO, the mortality of *D. suzukii* adults was not recorded, but a mortality of 100% was achieved for the concentrations of 5% and 10%. Scots pine oil caused a mortality rate of 82.5% for the concentration of 5% and 97.5% for the concentration of 10%, but for the lowest one, it had the effect of the 5%. The bergamot essential oil did not show an effect on the mortality of both the male and female flies. There was a statistically significant difference between both controls and all three applied concentrations (*p* < 0.05) for geranium EO and between the concentration of 1% and 10% *(p* < 0.05). A significant difference was observed between both controls and the 5% and 10% concentrations, between the 1% and 5% concentrations, as well as between the 1% and 10% concentrations for Scots pine (*p* < 0.05). There was also a significant difference between the concentrations of 5% and 10%. For dill oil, a significant difference was observed between both controls and the 5% and 10% concentrations and between the 1% and 5% and 1% and 10% concentrations (*p* < 0.05). No significant difference was observed for bergamot oil. Considering sex, in the contact test, females were slightly more resistant to treatments compared with males, as slightly less female flies died compared with the males (Table 2).

#### Emergence of the Adults after Exposure to Contact Toxicity Bioassay

During the assessment of the oviposition when exposed to contact treatment, for geranium EO, it was observed that there was no emergence of adults in all three concentrations (Table 3). A similar effect was obtained for Scots pine EO. For the lowest applied concentration (1%), it was noticed that there was an emergence of adults but compared with the control at a significantly lower level. For the concentration of 5% and 10% for dill EO, the emergence of adults was low or completely absent, but at the 1% concentration, there was a notable emergence of adults. No effect on adult emergence was recorded for bergamot EO. The difference between the sex of newly emerged flies was not significant (Table 4).

There was a statistically significant difference between both controls and all three applied concentrations (*p* < 0.05) for geranium, dill, and Scots pine.

### 2.3. Repellent Test Results

In the repellent test, the flies could choose between an untreated cotton wick and one treated with essential oil, and their position in the Y-tube olfactometer was recorded (Table 5). Geranium, dill, and Scots pine tested EOs notably repelled the flies after 24 h for both sexes. The highest repellency was achieved by geranium EO, with a preference index of −0.8, −1, and −1 for all three applied concentrations, respectively. The pine EO’s highest repellency was shown at the concentration of 1% (PI = −1) but also achieved very good repellency at the other two concentrations (PI = −0.6 and −0.8, respectively). Similar results were obtained for dill EO; the concentration of 1% PI was −0.2, and for the 5% and 10% PI, it was −0.6. For the bergamot oil, there was no record of consistent activity. Only bergamot EO showed an attractant effect for the concentration of 5% and a neutral effect after 24 h for the concentration of 10%. Comparing sex differences in response to the repellent test (Appendix A), there was no significant difference in the test for geranium and dill EOs. For Scots pine EO, it was noticed that it was more of an attractant for *D. suzukii* females than for males, while for the bergamot EO we obtained the opposite effect, which indicated that this oil was more of an attractant for males. Geranium, dill, and Scots pine EOs notably repelled the flies after 24 h for both sexes.

### 2.4. Multiple-Choice Bioassay

Geranium EO showed deterrent activity in all three applied concentrations, even after the fourth day (Table 6). The number of eggs laid on the berries treated with 1% and 5% EOs solutions (≤=3) was significantly lower than the control treatments. A low number off-laid eggs was observed on berries treated at a concentration of 5%, but there was no record of evidence of adult emergence. For the lowest applied concentration, there was observed adult emergence (Table 7). Only one egg after the second day was recorded for the highest applied concentration. Adult emergence was not recorded at all. Very similar results were obtained for the dill EO. A low number of eggs was laid for all three concentration compared with the controls. For the concentrations of 5% and 10%, the number of laid eggs over four days was low (≤0.25). Scots pine oil had a moderate effect compared with geranium and dill Eos, and its deterrent activity significantly decreased after the second day. After the first day, the number of laid eggs was similar to the geranium and dill EOs for all three concentrations. A slight increase in the number of laid eggs was observed after the second day compared with the geranium and dill EOs, but a significant rate of oviposition was observed after the third and fourth day. The number of laid eggs was ≤0.25 only for the highest applied concentration (10%). The emergence rate is shown in Table 8. The difference between the sex of newly emerged flies in the treatment was not significant (Table 9). Egg production also varied on age, genetic difference, the overall health of a female fly, or previous mating of adults.

For the first, second, third, and fourth day, there was a statistically significant difference between both controls and all three applied concentrations (*p* < 0.05) for geranium and dill. There was a statistically significant difference between both controls and all three applied concentrations (*p* < 0.05) for the first, second, and third day for Scots pine. After the third day, there was a significant difference between the 1% and 10% concentrations. After the fourth day, there was no significant difference between the untreated control and the lowest concentration (1%), but there was a significant difference between the 1% and 5% and between the 1% and 10% concentrations.

For the first, second, third, and fourth day, there was a statistically significant difference between both controls and all three applied concentrations (*p* < 0.05) for geranium, dill, and Scots pine. There was a significant difference between the 1% and 5% and between 1% and 10% concentrations after the fourth day for Scots pine oil.

### 2.5. EOs Chemical Composition

The chemical composition of essential oils for three EOs is presented in Table 9. The chromatographic analysis identified 46 different chemical compounds from the 3 analyzed EOs (Table 10). The obtained results showed that the most identified compounds belong to monoterpenes, oxygenated monoterpenes, and sesquiterpenes. Twenty-two constituents were identified in geranium essential oil, which comprised 97.68% of the total oil. The most abundant compounds were citronellol (22.16%), geraniol (21.22%), dihydrocitronellol acetate (13.83%), and geranyl acetate (8.88%). In the Scots pine EO, twelve compounds were identified. Isobornyl acetate was the main compound of this oil (84.52%).

Different from the previous two oil samples, in which compounds from the group of oxygenated monoterpenes were dominant, in dill EO, monoterpenes hydrocarbons were more represented (67.03%) compared with oxygenated monoterpenes (31.94%). The main compounds were limonene (44.53%), carvone (30.11), and α-phellandrene (11.45%).

## 3. Discussion

This study presents the insecticidal effects of four EOs on the males and females of *D. suzukii*. The chemical profiles of three promising EOs were characterized qualitatively using the GC/MS technique. Bergamot EO was not included in the chemical analysis as it did not show significant efficacy on the flies. EOs are complex mixtures, and their chemical composition and proportions of the main substances may vary depending on many elements, including extraction method, geographic origin, environmental factors, harvesting time, the plant part used for the extraction, plant species, and botanical variety, which directly affects their bioactivity. The use of EOs as bioinsecticides is of great interest as they can be used in organic and conventional agriculture production with little or no negative effect on the environment. As EOs can have different insecticidal effects [34], the aim of this study was to evaluate the different effects of oils through fumigant and contact toxicity essays and also evaluate their deterrent, repellent, or attractant effects on the adults of *D. suzukii*, which will be of great interest for further research using these oils for this important fruit pest. In this analysis, the most frequent compounds found can be grouped into three main categories (monoterpene hydrocarbons, oxygenated monoterpenes, and sesquiterpenes).

*P. graveolens* (*P. asperum*) essential oil, also known as geranium essential oil, has been reported to have insecticidal properties and repellent effects against various insects. *P. graveolens* and its constituents have been reported to have insecticidal activity against maize weevil (*Sitophilus zeamais* Motschulsky) [45], *Rhyzopertha dominica* Fabricius [46], Japanese termite (*Reticulitermes speratus* Kolbe) [47], the sweet potato whitefly (*Bemisia tabaci* Gennadius) [48], the house fly (*Musca domestica* L.) [49], etc. Geranium essential oil was highly repellent in our study, which indicated its prominent potential as a repellent for *D. suzukii*. Geranium oil has a repellent effect on females and males as previously reported in choice and no-choice bioassays; this is in accordance with this study as well, as the oil showed repellent and deterrent effects in the multiple-choice and repellent tests [50,51]. In similar tests, also regarding the assessment of the repellent effects, geranium (*P. asperum*) was reported to be a great source of repellent for this pest [25], which is in accordance with this study. Considering geranium’s complex chemical composition, it may act on more than one site, but EOs mostly affect the central nervous system, causing paralysis and death of the insects [52]. The identified major compounds found in our study were citronellol (22.16%) and geraniol (21.22%), which are most likely responsible for the insecticidal activity against *D. suzukii* as well [35,52]. These components in the geranium essential oils might play a primary role in the mode of action against *D. suzukii*. Some chemical compounds present in smaller amounts in the mixture of essential oils may act as synergists with major compounds for the expression of the insecticidal effect. Our study showed that the EOs of *P. graveolens* were toxic to *D. suzukii* adults in contact and fumigant-contact tests, but there is no available data from the literature on these tests, which make this study the first one for laboratory experiments with respect to geranium EO assessment. Geranium EO showed higher efficacy in the lowest concentration applied in four tests compared with the other tested EOs. This oil seems to be a promising nominee for further testing in the control of *D. suzukii* flies. The deterrent and repellent effect of EOs provides a future perspective for the use of EOs as active compounds for the field trials, as olfactory cues are important elements for the oviposition site choice of *D. suzukii* females, which will contribute to sustainable pest control in integrated pest management and organic growing as well.

*P. sylvestris* EO was evaluated in the fumigant test using a glass cylinder and in the contact test using a topical application, where the EOs were applied using a micro syringe administered to the ventral abdomen of adults. In study [53], among other 21 tested EOs in the concentration range of 2.94–11.76 mg/L for the fumigant test and in the concentration range of 1.25–20 µg dissolved in acetone (1 µL) for the contact test, *P. sylvestris* showed the highest efficacy. In the smaller amount applied (1%) *P. sylvestris* did not have a sufficient effect in our study as well in both the fumigant-contact and contact tests. The effect was achieved at the higher concentrations applied (10%), but it showed a good effect in the repellent and oviposition deterrent tests. Keeping in mind that *D. suzukii* may feed on the Pinus spp. tree honeydew as a possible food source, we expected the opposite result, that is, an attractant effect. Isobornyl acetate was the most present compound in the *P. sylvestris* oil in our study, having a percentage of 84.52%. Isobornyl acetate is a precursor in the pathway of camphor [54], which acts as a repellent against stored-product pests and mosquitos [55,56]. Comparably, the compound isobornyl acetate may act as a repellent, but there are no available literature studies regarding *P. silvestris* EO’s effect on adults of *D. suzukii*, which suggests the further testing of this essential oil as a deterrent and repellent against *D. suzukii* flies.

Dill (*A. graveolens*) EO’s main constituents were limonene (44.53%), carvone (30.11%), α-phellandrene (11.45%), terpinolene (4.97%), myrcene (1.82%), and α-pinene (1.77%). Dill essential oil has been reported as the most potent repellent against adults of German cockroach (*Blattella germanica* L.) [57]. This oil also has larvicide, pupicide, and oviposition deterrent effects on the dengue fever mosquito, *Aedes aegypti* Linn [58]. Similar to *P. sylvestris* oil, it showed a low effect for the lowest applied concentrations (1%) but high mortality and repellency for the higher applied concentrations (5% and 10%). Bedini et al. (2020) [59] reported that the EO of mandarin fruit (*Citrus reticulata* Blanco) showed repellent activity in a two-choice bioassay olfactometer and a deterrent effect in a two-choice cage test where a recording of laid eggs on mock fruits was performed. Limonene was the most dominant compound found in the oil, which may have the highest influence in these two tests. Also, limonene was the main dominant compound in the dill EO in our study as well, but in a much lower range compared with mandarin fruit. However, the deterrent effect of dill oil may be connected to this compound.

Essential oils are gaining more importance for the control of *D. suzukii*, which is why research on this topic has increased in recent years. Twelve different EOs were studied as potential repellents for *D. suzukii*; the best effect was achieved with peppermint oil, while high male mortality was achieved with thyme oil [50]. EOs of *Citrus reticulata, Melaleuca alternifolia, Cymbopogon winterianus,* and *Thymus vulgaris* also act as repellents against *D. suzukii* [25,59]. Species from *Cymbopogon* spp. and *Mentha* spp. EOs show high toxicity in topical application assays, while EOs of *C. verum* and *C. citratus* have high toxicity in different types of bioassays through ingestion and the reduction of oviposition [52]. EOs of *Eucalyptus citriodora* and *Melaleuca teretifolia* belonging to the Myrtace family show potential as a fumigant and have contact toxicities against this pest [60]. The EO of *Mentha arvensis* L. reduces the emergence of *D. suzukii* [61]. EOs of *P. aduncum, P. gaudichaudianum*, and *P. marginatum* also affected oviposition and had an impact on mortality in bioassays conducted using ingestion and topical application [62].

Comparing the four tested oils, it can be concluded that bergamot EO should be excluded from further testing. In the fumigant-contact and contact tests with higher applied dose levels, the other three oils were able to determine a high level of mortality, ranging from 92.5% to 100%. In the bioassays of repellent compounds, these oils showed a repellent effect after 24 h. The finding that dill and Scots pine EOs can firstly be attractive to flies can be considered interesting and needs to be studied further as it can be an aspect in attract and kill strategies. Geranium and dill EOs showed a high deterrent effect, even at the lowest applied doses (1% and 5%), with a mean of laid eggs being less than one compared with Scots pine, which showed a high level of deterrent effect at high doses (10%). When we compare the chemical composition of these three oils, we can affirm that their compositions are rather different; however, all three show insecticidal activity against *D. suzukii.* A good strategy for developing and formulating a bioinsecticide may be the combination of these oils as they probably have different modes of action on this pest.

In light of the above considerations, EOs are a big group of biologically active substances with insecticidal effects and a wide arsenal of different modes of action, which is especially important for delaying *D. suzukii* development or avoiding its resistance to insecticides.

## 4. Materials and Methods

### 4.1. Drosophila Suzukii Colony

The adults of *D. suzukii* used in these bioassays were obtained from a laboratory colony maintained from 2020 at the Department of Plant and Environmental Protection, Faculty of Agriculture, University of Novi Sad, Serbia, and from the Insectarium facility of CIHEAM Bari, Italy. Adults in the Insectarium of CIHEAM were kept in Plexiglas cages under controlled conditions (22 °C ± 1 °C; 62% ± 4%, 12:12 h (light:dark) with cornmeal medium. The colony in Serbia was kept under control conditions (23 °C ± 1 °C, 65% ± 5% relative humidity (RH), and photoperiod of 12:12 h (light:dark). The SWD laboratory colony is supplemented once a year with wild-caught adults. Specimens of *D. suzukii* were bred in glass jars (89 mm in diameter × 140 mm in height) containing an artificial diet in a small Petri dish (60 mm in diameter × 15 mm in height) based on corn flour, sugar, and yeast, following the methodology proposed by Schlesener et al. (2018) [63]. The experiment was carried out by using organic blueberry fruits purchased from the supermarket. This fruit was chosen because it is available throughout the year. We used 4–7-day-old adults (males and females) for the bioassays.

### 4.2. Essential Oils

The essential oils (EOs) used in this bioassay were geranium essential oil (*Pelargonium graveolens* L’Hér.), dill weed essential oil (*Anethum graveolens* L.), and Scots pine (*Pinus sylvestris* L.) oil, obtained from Avena Lab—Farmadria © (Vršac, Serbia). Bergamot essential oil (*Citrus bergamia* L.) was obtained from Eterra—Chemmax© doo (Novi Sad, Serbia). EOs were dissolved in acetone at three different concentrations (1%, 5%, and 10%). The first concentration had approximately 250 µL dissolved in 25 mL of pure acetone (1%), the second had 1250 µL (5%), and the third had 2500 µL dissolved in acetone (10%). As a control, water (Control 1) and acetone (Control 2) were applied.

### 4.3. Fumigant-Contact Toxicity

The experimental part was conducted at the Department of Plant and Environmental Protection, Faculty of Agriculture, University of Novi Sad, Serbia. A metal ring (9 cm in height, 8.5 cm inner diameter) on the bottom part of the Petri dish (8.8 inner diameter) was employed for the fumigant-contact toxicity assays. A heavy glass cover was placed on the top of the ring to prevent gas from leaking. EOs dissolved in acetone were applied to a paper disc. After 20 min of evaporating acetone, the paper disc was placed on the bottom part of the Petri dish. Ten adult SWDs (5 males and 5 females) were placed in the Petri dish. A cotton wick soaked with 10% sugar solution was also placed inside the ring to feed the adults. The test containers were maintained in the chamber room at 23 °C ± 1 °C, 65% ± 5% RH and using a 12 h:12 h light:dark cycle. After 24 h of setting the experiment conditions, the flies were transferred to a new Petri dish and then counted. The adult flies were considered dead if their appendages did not move after being touched with a fine brush. All treatments were replicated 4 times. This is a combined test (fumigant-contact) as the flies are not physically separated from the treated surface.

### 4.4. Contact Toxicity

The experimental part was conducted at the Integrated Pest Management Laboratory of CIHEAM Bari. Transparent polystyrene containers (50 mL) were used as experimental units. The experiment was performed with some modifications to the previously reported method [64]. EOs were sprayed using a spray finger onto the bottoms and sides of the experimental containers. An opening was covered with fine mesh with the aim of avoiding saturation with the EOs volatiles while being thick enough to prevent flies from escaping. Paper discs were placed on the bottom of the containers. Before releasing insects into the containers, the spray cover needed to evaporate to dryness. Ten *D. suzukii* flies (5 males and 5 females) were placed in each test container. All three concentrations (1, 5, and 10%) of each EO were tested in four replicates (n = 4) in addition to control treatments. Test containers were kept under control conditions (22 °C ± 2 °C, 62% ± 4% RH, 12 h:12 h light:dark cycle). Control containers were sprayed with water and acetone, and left to complete dryness. The mortality of the flies when also considering sex was recorded after an exposure time of 24 h. In the container, small wet tissue and a small amount of medium were placed provide a water supply and feed insect adults and to also follow females’ ability to lay eggs in a stressful environment. Newly hatched flies began to be counted after ten days of the experiment setting. Flies were counted for a few days until three days in a row where no newly hatched flies were found. The sex of newly emerged flies was also recorded.

### 4.5. Repellent Effect

The experimental part was conducted at the Department of Plant and Environmental Protection, Faculty of Agriculture, University of Novi Sad, Serbia. The repellent effect of EO volatiles was evaluated using a glass Y-tube olfactometer consisting of two arms (10 cm long by 2.5 cm internal diameter). The assay was slightly modified from the previously reported assay [65]. Cotton wool soaked with 1 mL of EOs dissolved in acetone was placed on the left side of the olfactometer, and a control treatment soaked in acetone was placed on the right side. A cotton wick soaked in a solution of sugar and water was placed in the central tube of the olfactometer and attached using electrical tape to keep it in place. Ten insects (5 males and 5 females) were inserted into the central tube of the olfactometer, after which the entrance was closed with a fine mesh. All openings on the olfactometer were closed with a fine mesh to enable gas exchange. The effect was observed after half an hour, 2 h, 4 h, 12 h, and 24 h. The position of flies was recorded (number of flies in test arm, number of flies in control arm and in the central tube) after a certain time of exposure, and a preference index was calculated. Flies in the central tube were calculated in the control. Also, the movement of flies in relation to sex was monitored to see the difference in response between males and females.
$$\mathrm{Preference}\;\mathrm{index}=\frac{\mathrm{number}\;\mathrm{of}\;\mathrm{flies}\;\mathrm{in}\;\mathrm{test}\;\mathrm{arm}\;-\;\mathrm{number}\;\mathrm{of}\;\mathrm{flies}\;\mathrm{in}\;\mathrm{control}\;\mathrm{arm}}{\mathrm{number}\;\mathrm{of}\;\mathrm{flies}\;\mathrm{in}\;\mathrm{test}\;\mathrm{arm}\;+\;\mathrm{number}\;\mathrm{of}\;\mathrm{flies}\;\mathrm{in}\;\mathrm{control}\;\mathrm{arm}}$$

Scale for preference index: from −1.00 to −0.10—repellent activity; from −0.10 to +0.10—neutral activity; from +0.10 to +1.00—attractant activity.

### 4.6. Multiple-Choice Test of Egg-Laying Activity

The experimental part was conducted at the Integrated Pest Management Laboratory of CIHEAM Bari. A multiple-choice test was conducted in a plastic arena (55 cm length, 36 cm width, and 16 cm depth) with 9 ventilation holes covered with fine mash on every side of the arena. The tenth hole was used as an entrance hole, through which flies were inserted, and then closed. The arena was covered with a heavy glass cover lid. In the arena, six small bottom parts of the Petri dish were placed and fixed with glue pads. In one Petri dish wet wipes are placed and a diet medium as well, as a source of food and water. Blueberries were used for the experiment. The experiment was conducted in four replicates. On the first day, all blueberries needed for the experiment were submerged for two seconds in three different essential oil solutions and allowed to dry completely. Blueberries for the control treatment were submerged for two seconds in acetone and water. After drying, one blueberry was placed in a Petri dish separately. The rest of the berries, which would be used in the following days, were placed in a container, covered with fine mesh, and left in the same chamber room as the experimental units. The aim was to treat all berries only on the first day in order to assess the EOs’ effect after 24, 48, 72, and 96 h. After placing berries in the arena through the entrance hole, 10 flies were inserted (5 males and 5 females). After 24 h, flies were aspirated, and egg counting on the berries was conducted using a stereomicroscope. Arenas were kept in a climate chamber room under control conditions (22 °C ± 2 °C, 62% ± 4% RH, 12 h:12 h light:dark cycle). After counting eggs, every berry was placed in a separate plastic cup, covered with fine mesh, and placed with a small amount of sand to prevent berries from spoiling with the aim of causing adults’ emergence. The cups were placed in a chamber room in order to record the emergence of adults. New treated berries were placed as described above for four days in total in order to understand how long the essential oil being applied to the blueberries exerted a deterrent effect for *D. suzukii* female oviposition.

### 4.7. Statistical Analysis

Statistical software Statistica version 14.0.0.15 (Tibco Software Inc., Palo Alto, CA, USA, 2021) was used for the statistical analysis in this study. An analysis of variance (one-way ANOVA) test was used to test the mortality of EOs in different concentrations compared with the untreated control and the control treated with only acetone. The difference between treatments was compared using a Duncan post hoc test (Duncan’s multiple range test). Duncan’s multiple range test is based on the comparison of the range of a subset of the means with a calculated significant range [66]. The chi-square test was used to determine whether there was statistical significance between sex in mortality in the fumigant-contact, contact, and multiple-choice tests. The chi-square test of independence (Pearson chi-square test) is one of the most useful statistical tests when the variables are nominal [67]. This test is useful when it is necessary to determine whether any obtained frequencies deviate from the frequencies we would expect under a certain hypothesis.

### 4.8. GC–MS Analysis

For the chemical characterization of essential oils, gas chromatography with mass spectrometry (GC–MS Shimadzu QP2010) was used. The GC conditions were as follows: fused silica HP-5 column; carrier gas He (1.0 mL/min); temperature programmed from 55 °C to 240 °C with a temperature increase of 3 °C/min; injection port temperature 250 °C; and detector temperature 280 °C. Ionization of the sample components was performed in the EI mode (70 eV). A mixture of n-alkanes (C8-C40) was injected under the above conditions to calculate the retention indices using the generalized equation [68]. The retention indices of n-alkanes were used for recalculating the retention indices of volatile constituents. The identification of volatile constituents was performed by comparing their retention indices and MS spectra with those presented in the databases available in the licensed MassFinder 4 software (EssentialOil 4a and Adams2205 databases [44]). Oil samples were dissolved in n-hexane prior to GC–MS analysis.

## 5. Conclusions

To the best of our knowledge, regarding *P. silvestris*, *C. bergamia,* and *A. graveolens* EO repellent and deterrent activity against adults of *D. suzukii*, this is the first report on the use of these three EOs against *D. suzukii*. The essential oil of geranium (*A. graveolens*) showed the highest insecticidal effect in all four applied bioassays on *D. suzukii*, even at the lowest applied concentration. Dill and Scots pine EOs have more importance as repellents and oviposition deterrents, even at the lower applied concentration, which should be taken into account during future trials. Further research will be needed in order to detect the EC_50_ of these EOs excluding bergamot EO as it did not show any promising insecticidal effect (effect on mortality, emergence, repellency, or on oviposition deterrence for *D. suzukii*). Plant EOs and/or their components could be an efficient alternative to chemical insecticides. EOs can be a powerful and prospective tool in organic agriculture, but it would be necessary to evaluate the side effects of EOs for the natural enemies used for the management of *D. suzukii* population. Organoleptic properties should be also taken into consideration, which can affect market prosperity. Taking into account that these are photosensitive and rapidly degrading compounds, it is necessary to find a suitable method of formulation that contains essential oil as an active substance.

## Figures and Tables

**Table 1 plants-12-03727-t001:** Mortality of adult flies in the fumigant-contact and contact bioassays.

Mortality (%) ± SD in Fumigant-Contact Bioassay				
Treatments	Geranium	Bergamot	Dill	Scots Pine
1%	95 ± 0.57	2.50 ± 0.50	37.50 ± 1.70	32.50 ± 0.50
5%	97.50 ± 0.50	7.50 ± 0.95	97.50 ± 0.50	92.50± 1.50
10%	100 ± 0	5 ± 0.57	100 ± 0	100 ± 0
Control 1	2.50 ± 0.50	2.50 ± 0.50	2.50 ± 0.50	0 ± 0
Control 2	0 ± 0	2.50 ± 0.50	0 ± 0	0 ± 0
F-test	667.95	0.50	141.03	189.75
*p*-value	0.00	0.73	0.00	0.00
Mortality (%) ± SD in contact bioassay				
Treatments	Geranium	Bergamot	Dill	Scots pine
1%	87.50 ± 0.50	0 ± 0	0 ± 0	5 ± 0.57
5%	92.50 ± 1.50	0 ± 0	100 ± 0	82.50 ± 0.50
10%	100 ± 0	5 ± 0.57	100 ± 0	97.50 ± 0.50
Control 1	0 ± 0	2.50 ± 0.50	2.50 ± 0.50	0 ± 0
Control 2	0 ± 0	0 ± 0	0 ± 0	2.50 ± 0.50
F-test	210.65	1.71	2361	429.80
*p*-value	0.00	0.19	0.00	0.00

Mortality is calculated as the number of dead flies/40 × 100; SD—standard deviation.

**Table 2 plants-12-03727-t002:** Sex differences in mortality in the fumigant-contact and contact bioassay.

SUM of Dead Flies ± SD Considering Sex in Fumigant-Contact Bioassay								
	Geranium		Bergamot		Dill		Scots Pine	
Treatments	F	M	F	M	F	M	F	M
1%	19 ± 0.5	19 ± 0.5	0 ± 0	1 ± 0.5	6 ± 1	9 ± 0.95	7 ± 0.5	6 ± 0.57
5%	19 ± 0.5	20 ± 0	2 ± 0.57	1 ± 0.5	19 ± 0.5	20 ± 0	17 ± 1.5	20 ± 0
10%	20 ± 0	20 ± 0	1 ± 0.5	1 ± 0.5	20 ± 0	20 ± 0	20 ± 0	20 ± 0
Control 1	0 ± 0	1 ± 0.5	0 ± 0	1 ± 0.5	1 ± 0.5	0 ± 0	0 ± 0	0 ± 0
Control 2	0 ± 0	0 ± 0	1 ± 0.5	0 ± 0	0 ± 0	0 ± 0	0 ± 0	0 ± 0
χ^2^	155.79		39.70		237.14		166.90	
df	50		36		106		64	
*p*-value	0.00		0.30		0.00		0.00	
SUM of Dead Flies ± SD Considering Sex in Contact Bioassay								
	Geranium		Bergamot		Dill		Scots pine	
Treatments	F	M	F	M	F	M	F	M
1%	16 ± 0	19 ± 0.5	0 ± 0	0 ± 0	0 ± 0	0 ± 0	0 ± 0	2 ± 0.57
5%	18 ± 1	19 ± 0.5	0 ± 0	0 ± 0	20 ± 0	20 ± 0	16 ± 0	17 ± 0.5
10%	20 ± 0	20 ± 0	1 ± 0.5	1 ± 0.5	20 ± 0	20 ± 0	19 ± 0.5	20 ± 0
Control 1	0 ± 0	0 ± 0	0 ± 0	1 ± 0.5	0 ± 0	1 ± 0.5	0 ± 0	0 ± 0
Control 2	0 ± 0	0 ± 0	0 ± 0	0 ± 0	0 ± 0	0 ± 0	1 ± 0.5	0 ± 0
χ^2^	113.81		67.77		120.67		133.14	
df	40		22		22		22	
*p*-value	0.00		0.00		0.00		0.00	

χ^2^—chi-square test; df—degrees of freedom; SUM of flies considering sex; F—female; M—male; SD—standard deviation.

**Table 3 plants-12-03727-t003:** Mean number of emerged adults after exposure to contact toxicity bioassay.

$\overset{-}{x}$ ± SD				
Treatments	Geranium	Bergamot	Dill	Scots Pine
1%	0.25 ± 0.50	20.50 ± 1.29	7.75 ± 3.59	1 ± 0.81
5%	0.25 ± 0.50	19.25 ± 2.87	0.75 ± 1.50	0.25 ± 0.50
10%	0 ± 0	17 ± 1.82	0 ± 0	0 ± 0
Control 1	23.25 ± 7.18	20.75 ± 2.75	18.75± 3.77	23.25 ± 2.98
Control 2	23.00 ± 3.82	25.50 ± 2.38	19.50 ± 4.35	20.50 ± 1.91
F-test	47.38	7.32	36.54	206.25
*p*-value	0.00	0.00	0.00	0.00

Mean—average of a set of values; SD—standard deviation.

**Table 4 plants-12-03727-t004:** Sex differences in the emergence of the adults.

SUM of Emerged Flies ± SD Considering Sex in Contact Bioassay								
	Geranium		Bergamot		Dill		Scots Pine	
Treatments	F	M	F	M	F	M	F	M
1%	0 ± 0	1 ± 0.5	37 ± 2.36	45 ± 3.3	13 ± 0.95	18 ± 2.08	0 ± 0	4 ± 0.81
5%	0 ± 0	1 ± 0.5	45 ± 1.7	32 ± 1.63	0 ± 0	0 ± 0	1 ± 0.5	0 ± 0
10%	0 ± 0	0 ± 0	30 ± 3.1	38 ± 2.64	0 ± 0	0 ± 0	0 ± 0	0 ± 0
Control 1	42 ± 4.59	51 ± 3.3	41 ± 2.06	42 ± 3.82	37 ± 3.3	38 ± 1.29	53 ± 2.62	40 ± 0.81
Control 2	50 ± 3.1	42 ± 3.1	48 ± 2.58	54 ± 1.73	50 ± 3.87	28 ± 3.16	39 ± 0.95	43 ± 2.21
χ^2^	178.33		162.95		206.58		181.73	
df	120		162		148		134	
*p*-value	0.00		0.46		0.00		0.00	

χ^2^—chi-square test; df—degrees of freedom; SUM of emerged flies considering sex; F—female; M—male; SD—standard deviation.

**Table 5 plants-12-03727-t005:** Repellent activity of essential oils using a Y-olfactometer.

Treatments	Exposure Time				
Geranium	30 min	2 h	4 h	12 h	24 h
1%	−0.2 R	0.4 N	0 N	−0.8 R	−0.8 R
5%	0 N	−0.4 R	−0.8 R	−0.6 R	−1 R
10%	−0.4 R	−0.4 R	−0.4 R	−1 R	−1 R
Bergamot	Exposure time				
1%	0.2 A	0.2 A	0 N	−0.2 R	−0.2 R
5%	0 N	0 N	0.2 A	0.4 A	0.4 A
10%	0.4 A	0 N	0.2 A	0 N	0 N
Dill	Exposure time				
1%	−0.4 R	−0.6 R	−0.4 R	−0.2 R	−0.2 R
5%	0.2 A	0.2 A	−0.6 R	−0.6 R	−0.6 R
10%	−0.8 R	−0.6 R	−0.6 R	−0.8 R	−0.6 R
Scots pine	Exposure time				
1%	−0.6 R	−1 R	−1 R	−0.8 R	−1 R
5%	−0.4 R	−0.4 R	−0.6 R	−0.4 R	−0.6 R
10%	0.2 A	−0.4 R	−0.4 R	−1 R	−0.8 R

A—attractive; R—repellent, N—neutral.

**Table 6 plants-12-03727-t006:** Multiple-choice bioassay—mean number of laid eggs.

Mean ± SD					
Geranium	1st Day	2nd Day	3rd Day	4th Day	Mean ± SD
1%	0.50 ± 0.57	0.50 ± 0.57	0.75 ± 0.95	0.50 ± 0.57	2.25 ± 0.5
5%	0.25 ± 0.50	0.25 ± 0.50	0.25 ± 0.50	0 ± 0	0.75 ± 0.5
10%	0 ± 0	0.25 ± 0.50	0 ± 0	0 ± 0	0.25 ± 0.5
Control 1	8 ± 1.82	6.5 ± 3.00	7.25 ± 2.50	7.± 1.82	28.75 ± 2.5
Control 2	7.75 ± 2.98	7 ± 0.81	8.25 ± 1.70	6.25 ± 1.50	29.25 ± 3.5
F-test	27.24	23.60	32.32	42.67	/
*p*-value	0.00	0.00	0.00	0.00	/
Dill	1st day	2nd day	3rd day	4th day	
1%	0.75 ± 0.95	0.75 ± 0.95	1 ± 0.81	0.25 ± 0.50	2.75 ± 1.25
5%	0.25 ± 0.50	0 ± 0	0 ± 0	0 ± 0	0.25 ± 0.50
10%	0.25 ± 0.50	0.25 ± 0.50	0 ± 0	0 ± 0	0.5 ± 0.50
Control 1	6.75 ± 1.50	6.0 ± 1.41	5.50 ± 1.73	5.25 ± 1.25	23.5 ± 2.64
Control 2	5.25 ± 0.95	5.25 ± 0.50	4.50 ± 1.29	5.0 ± 2.16	20 ± 1.41
F-test	42.21	50.01	25.59	23.51	/
*p*-value	0.00	0.00	0.00	0.00	/
Scots pine	1st day	2nd day	3rd day	4th day	
1%	0.50 ± 0.57	1.25 ± 0.95	2.75 ± 0.95	6.25 ± 0.95	10.75 ± 10.2
5%	0.25 ± 0.50	0.75 ± 0.50	1.50 ± 0.57	2.25 ± 0.50	4.75 ± 3.5
10%	0 ± 0	0 ± 0	0 ± 0	0.25 ± 0.50	0.25 ± 0.50
Control 1	9.25 ± 2.36	8.50 ± 0.57	6.75 ± 1.50	7.0 ± 1.63	31.5 ± 4.79
Control 2	8.25 ± 1.50	8.25 ± 2.06	8.75 ± 2.21	8.75 ± 0.95	34 ± 1.15
F-test	51.87	62.71	32.03	49.7	/
*p*-value	0.00	0.00	0.00	0.00	/

Mean—average of a set of values; SD—standard deviation.

**Table 7 plants-12-03727-t007:** Multiple-choice bioassay of the mean number of emerged adults.

		Mean ± St. Dev.		
Geranium EO	1st day	2nd day	3rd day	4th day
1%	0.25 ± 0.50	0.50 ± 0.57	0.50 ± 0.57	0.25 ± 0.50
5%	0 ± 0	0 ± 0	0 ± 0	0 ± 0
10%	0 ± 0	0 ± 0	0 ± 0	0 ± 0
Control 1	6 ± 2.94	4.50 ± 3.0	6.75 ± 1.70	6.0 ± 1.41
Control 2	6.25 ± 2.98	6.25 ± 0.95	7.0 ± 1.63	5.50 ± 1.91
F-test	12.30	16.70	45.80	32.7
*p*-value	0.00	0.00	0.00	0.00
Dill	1st day	2nd day	3rd day	4th day
1%	0.50 ± 0.57	0.25 ± 0.5	1 ± 0.81	0 ± 0
5%	0 ± 0	0 ± 0	0 ± 0	0 ± 0
10%	0.25 ± 0.50	0 ± 0	0 ± 0	0 ± 0
Control 1	5.25 ± 2.06	4.50 ± 1.73	4.25 ± 1.50	4.75 ± 1.50
Control 2	4.75 ± 2.50	4.25 ± 1.50	4.50 ± 1.70	4.50 ± 2.51
F-test	12.32	30.69	21.06	14.97
*p*-value	0.00	0.00	0.00	0.00
Scots pine	1st day	2nd day	3rd day	4th day
1%	0 ± 0	1.25 ± 0.95	1.75 ± 0.95	3.75 ± 0.50
5%	0 ± 0	0.75 ± 0.50	0.75 ± 0.50	1.25 ± 0.50
10%	0 ± 0	0 ± 0	0 ± 0	0 ± 0
Control 1	7.50 ± 2.08	8.50 ± 0.57	5.50 ± 1.29	6.25 ± 2.21
Control 2	7.50 ± 1.29	8.25 ± 2.06	7.50 ± 1.91	7.50 ± 0.57
F-test	56.25	62.71	32.36	35.32
*p*-value	0.00	0.00	0.00	0.00

**Table 8 plants-12-03727-t008:** Emergence rate in the multiple-choice test.

	Emergence Rate %			
Geranium EO	1st day	2nd day	3rd day	4th day
1%	50	100	66.66	50
5%	0	0	0	0
10%	0	0	0	0
Control 1	75	73.07	93.10	84.71
Control 2	80.64	89.28	84.84	88
Dill	1st day	2nd day	3rd day	4th day
1%	66.66	33.33	100	0
5%	0	0	0	0
10%	50	0	0	0
Control 1	77.77	75	77.27	90.47
Control 2	90.47	80.95	88.88	85.71
Scots pine	1st day	2nd day	3rd day	4th day
1%	0	60	63.71	60
5%	0	0	50	55.55
10%	0	0	0	0
Control 1	81.08	88.23	81.48	89.28
Control 2	90.90	93.93	85.71	85.71

**Table 9 plants-12-03727-t009:** Sex differences in the emergence of adults in the multiple-choice test.

SUM of Emerged Flies ± SD Considering Sex in Multiple-Choice Bioassay								
	1st Day		2nd Day		3rd Day		4th Day	
Geranium EO	F	M	F	M	F	M	F	M
1%	1 ± 0.3	0 ± 0	2 ± 0.57	0 ± 0	1 ± 0.5	1 ± 0.5	0 ± 0	1 ± 0.5
5%	0 ± 0	0 ± 0	0 ± 0	0 ± 0	0 ± 0	0 ± 0	0 ± 0	0 ± 0
10%	0 ± 0	0 ± 0	0 ± 0	0 ± 0	0 ± 0	0 ± 0	0 ± 0	0 ± 0
Control 1	13 ± 1.67	11 ± 0.95	9 ± 1.50	10 ± 1.29	15 ± 0.95	12 ± 0.81	14 ± 1.73	10 ± 0.57
Control 2	14 ± 1.51	11 ± 1.70	13 ± 0.57	12 ± 0.81	15 ± 1.7	13 ± 1.7	13 ± 1.25	9 ± 0.95
χ^2^	127.77		127.77		114.61		158.41	
df	120		78		78		78	
*p*-value	0.00		0.00		0.00		0.00	
	1st day		2nd day		3rd day		4th day	
Dill EO	F	M	F	M	F	M	F	M
1%	1 ± 0.5	1 ± 0.5	1 ± 0.5	0 ± 0	2 ± 0.57	2 ± 0.57	0 ± 0	0 ± 0
5%	0 ± 0	0 ± 0	0 ± 0	0 ± 0	0 ± 0	0 ± 0	0 ± 0	0 ± 0
10%	0 ± 0	1 ± 0.5	0 ± 0	0 ± 0	0 ± 0	0 ± 0	0 ± 0	0 ± 0
Control 1	11 ± 1.5	10 ± 0.57	9 ± 0.95	9 ± 1.25	8 ± 0.5	9 ± 1.15	11 ± 2.21	8 ± 0.81
Control 2	9 ± 0.95	10 ± 1.73	15 ± 1.5	14 ± 1.0	9 ± 0.81	12 ± 1.07	7 ± 0.5	11 ± 2.5
χ^2^	166.26		177.72		138.08		160.00	
df	106		106		92		64	
*p*-value	0.00		0.00		0.00		0.00	
	1st day		2nd day		3rd day		4th day	
Scots pine EO	F	M	F	M	F	M	F	M
1%	0 ± 0	0 ± 0	2 ± 0.57	1 ± 0.5	5 ± 0.5	2 ± 0.57	8 ± 0.81	7 ± 0.5
5%	0 ± 0	0 ± 0	0 ± 0	0 ± 0	3 ± 0.5	0 ± 0	3 ± 0.5	2 ± 0.57
10%	0 ± 0	0 ± 0	0 ± 0	0 ± 0	0 ± 0	0 ± 0	0 ± 0	0 ± 0
Control 1	17 ± 0.95	13 ± 1.25	13 ± 1.5	17 ± 1.89	18 ± 0.95	11 ± 0.5	13 ± 1.25	12 ± 1.15
Control 2	14 ± 1.72	16 ± 1.15	17 ± 0.95	14 ± 1.29	11 ± 1.29	12 ± 0.81	17 ± 0.95	13 ± 0.95
χ^2^	147.85		167.14		174.54		168.09	
df	92		106		120		120	
*p*-value	0.00		0.00		0.00		0.00	

χ^2^—chi-square test; df—degrees freedom; SUM of emerged flies considering sex; F—female; M—male.

**Table 10 plants-12-03727-t010:** Chemical composition of *P. graveolens*, *P. sylvestris*, and *A. graveolens* essential oils (EOs).

Compound	RI_exp_	RI ^a,b^	Relative Content (%)			IM ^c^
			Geranium	Scots Pine	Dill	
*α*-Thujene	923	924 ^a^	/	/	0.18	RI ^a^, MS
*α*-Pinene	930	932 ^a^	1.64	1.73	1.77	RI ^a^, MS
Camphene	943	946 ^a^	/	4.4	/	RI ^a^, MS
*β*-Pinene	973	974 ^a^	/	0.11	0.49	RI ^a^, MS
Myrcene	988	988 ^a^	/	/	1.82	RI ^a^, MS
*α*-Phellandrene	1003	1002 ^a^	/	/	11.45	RI ^a^, MS
δ-3-Carene	1009	1008 ^a^	0.38	1.42	/	RI ^a^, MS
*o*-Cymene	1022	1022 ^a^	0.22	/	/	RI ^a^, MS
α-Terpinene	1014	1014 ^a^	/	/	1.23	RI ^a^, MS
Limonene	1026	1024 ^a^	2.77	0.68	44.53	RI ^a^, MS
γ-Terpinene	1056	1054 ^a^	/	/	0.59	RI ^a^, MS
Terpinolene	1086	1086 ^a^	/	/	4.97	RI ^a^, MS
Linalool	1100	1095 ^a^	1.02	/	0.26	RI ^a^, MS
cis-Rose oxide	1110	1106 ^a^	0.39	/	/	RI ^a^, MS
cis-β-Terpineol	1143	1140 ^a^	0.73	/	/	RI ^a^, MS
Menthone	1152	1148 ^a^	1.14	/	/	RI ^a^, MS
Isoborneol	1155	1155 ^a^	/	tr	/	RI ^a^, MS
*iso*-Pulegol	1163	1155 ^a^	0.58	/	/	RI ^a^, MS
Borneol	1164	1165 ^a^	/	0.44	/	RI ^a^, MS
Terpinen-4-ol	1176	1174 ^a^	/	/	0.1	RI ^a^, MS
*p*-Cymen-8-ol	1186	1179 ^a^	/	/	0.11	RI ^a^, MS
α-Terpineol	1190	1186 ^a^	6.03	/	tr	RI ^a^, MS
*ciss*-dihydro Carvone	1196	1191 ^a^	/	/	0.88	RI ^a^, MS
γ-Terpineol	1197	1199 ^a^	2.11	/	/	RI ^a^, MS
trans-dihydro Carvone	1204	1200 ^a^	/	/	0.31	RI ^a^, MS
Citronellol	1228	1223 ^a^	22.16	/	/	RI ^a^, MS
Fenchyl acetate	1232	1229 ^a^	/	tr	/	RI ^a^, MS
Carvone	1243	1239 ^a^	/	/	30.11	RI ^a^, MS
Geraniol	1254	1249 ^a^	21.22	/	/	RI ^a^, MS
Citronellyl formate	1275	1271 ^a^	0.63	/	/	RI ^a^, MS
Isobornyl acetate	1286	1283 ^a^	/	84.52	/	RI ^a^, MS
Geranyl formate	1302	1298 ^a^	0.95	/	/	RI ^a^, MS
Dihydro Citronellol acetate	1320	1319 ^a^	13.83	/	/	RI ^a^, MS
*iso*-dihydro Carveol acetate	1329	1326 ^a^	/	0.23	/	RI ^a^, MS
α-Terpinyl acetate	1348	1346 ^a^	/	/	0.1	RI ^a^, MS
Citronellyl acetate	1353	1350 ^a^	2.34	/	/	RI ^a^, MS
Neryl acetate	1364	1359 ^a^	4.79	/	/	RI ^a^, MS
Gerany acetate	1383	1379 ^a^	8.88	/	/	RI ^a^, MS
*α*-Gurjunene	1407	1409 ^a^	4.87	/	/	RI ^a^, MS
Caryophyllene	1417	1417 ^a^	/	0.16	/	RI ^a^, MS
*α*-Humulene	1451	1452 ^a^	/	0.1	/	RI ^a^, MS
*allo*-Aromadendrene	1458	1458 ^a^	0.74	-	/	RI ^a^, MS
γ-Gurjunene	1470	1475 ^a^	0.26	-	/	RI ^a^, MS
*α*-Zingiberene	1494	1943 ^b^	/	/	tr	RI^b^, MS
Dill apiole	1624	1620 ^a^	/	/	0.38	RI ^a^, MS
*p*-Camphorene	1951	1980 ^b^	/	/	tr	RI ^b^, MS
Monoterpene hydrocarbons			5.01	8.34	67.03	
Oxygenated monoterpenes			86.41	85.19	31.94	
Sesquiterpenes			5.87	0.26	-	
Other			0.39	-	0.38	
TOTAL			97.68	93.79	99.43	

t-trace (<0.1); RI ^a,b^—retention index from the literature; ^a^—Adams 2205 database [44]; ^b^—EssentialOil 4a database; RI—retention indices calculated from retention times in relation to those of a series of n-alkanes C8-C40 on a 30 m DB-5 capillary column; MS—mass spectra; IM ^c^—identification method was MS based on a comparison with the Adams 2205 and EssentialOil 4a databases.

## Data Availability

Not applicable.

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
