# Peer review of "Insecticidal Activity of Selected Essential Oils against Drosophila suzukii (Diptera: Drosophilidae)"

_plants, 2023, doi:10.3390/plants12213727_

Round 1

Reviewer 1 Report

The present paper is very well written, and the experiments that have been carried out are excellently planned and conducted.

I recommend a small correction of the English language, the text needs minor corrections.

In the abstract, the conclusion is not entirely convincing, because it is clear that only geranium oil shows serious activity.

lines 76 -77 ... the effects of EOs were evaluated on adults, males, and females. I hope it is D. suzukii, not human individuals.

I recommend a small correction of the English language, the text needs minor corrections.

Author Response

Reply to the reviewer 1:

Comment 1: The present paper is very well written, and the experiments that have been carried out are excellently planned and conducted.

Response: Thank you for your time, effort and kind words. We are grateful for the valuable comments and suggestions.

Comment 2: I recommend a small correction of the English language, the text needs minor corrections.

Response:  In addition to the above comments, all spelling and grammatical errors pointed out by the reviewers have been corrected.

Comment 3: In the abstract, the conclusion is not entirely convincing, because it is clear that only geranium oil shows serious activity.

Response: Thank you for pointing this out. The reviewer is correct, and we have updated the conclusion in the abstracts.

Lines 26-31: Scots pine and dill EOs have a moderate to strong effect in most tests. The mortality of 100% was achieved for the highest applied concentration in fumigant-contact and contact tests. Geranium, scots pine, and dill have great potential to be used as an environmentally friendly way of controlling D. suzukii, since they exhibit deterrent, repellent and insecticidal effects.

Comment 4: lines 76 -77 ... the effects of EOs were evaluated on adults, males, and females. I hope it is D. suzukii, not human individuals.

Response:  Lines 78-80: The revised text reads as follows on: effects of EOs were evaluated on adult flies, males and females.

Reviewer 2 Report

"Manuscript ' Insecticidal activity of selected essential oils against Drosophila suzukii (Diptera: Drosophilidae)' assessed the insecticidal effects of four different essential oils (Pelargonium graveolens, Anethum graveolens, Pinus sylvestris, and Citrus bergamia) on Drosophila suzukii in four distinct approaches and complemented the analysis with GC-MS of the constituents of these three essential oils. This study's findings will serve as a valuable reference for integrated pest management of Drosophila suzukii. However, the manuscript also contains numerous issues, particularly requiring substantial revisions in data analysis."

Table 1.

What do "Control 1" and "Control 2" refer to? Please provide a detailed description. Additionally, after performing the Duncan's Multiple Range test for data analysis, it is necessary to indicate the differences between the treatments instead of just providing a p-value. Similar modifications are required for data in other tables. Furthermore, all instances of "p" in the manuscript should be italicized.

Table 2.

Additional details regarding the fumigant-contact and contact tests should be provided to clarify which gender exhibited higher toxicity based on the existing data.

Table 4.

More detailed data should be presented, including the number of hatched females and males.

Table 6.

It is necessary to calculate the total egg production per female under different treatments. The significant difference in total egg production among the control groups in various essential oil treatments needs to be explained. For instance, in the Geranium experiment, Control 2 had a total egg production of 29.25, whereas in the Dill experiment, Control 2 had only 20. An explanation for this variation should be provided.

Table 7.

Calculating the emergence rate under different treatments may be more scientifically sound, considering the varying egg production among females. A direct comparison of emerged adults may not be meaningful. The same modification should be applied to Table 3.

Discussion.

The current discussion section needs significant revision. For instance, in lines 384-398, despite extensive discussion, the intended conclusions of the authors are not clear.

The results of the GC-MS analysis are crucial supporting data and of great interest to readers. A detailed analysis of which components in the essential oils might play a primary role should be included.

Please incorporate these revisions to enhance the clarity and scientific rigor of the manuscript.

Moderate editing of English language required

Author Response

Response to the reviewer 2:

Comment 1:"Manuscript ' Insecticidal activity of selected essential oils against Drosophila suzukii (Diptera: Drosophilidae)' assessed the insecticidal effects of four different essential oils (Pelargonium graveolens, Anethum graveolens, Pinus sylvestris, and Citrus bergamia) on Drosophila suzukii in four distinct approaches and complemented the analysis with GC-MS of the constituents of these three essential oils. This study's findings will serve as a valuable reference for integrated pest management of Drosophila suzukii. However, the manuscript also contains numerous issues, particularly requiring substantial revisions in data analysis."

Response: We would like to take this opportunity to thank you for your effort and contribution in reviewing our manuscript, without which it would be impossible to maintain the high standards of peer-reviewed journals. We will address all your concerns in the revised version of the manuscript.

Comment 2: Table 1.

What do "Control 1" and "Control 2" refer to? Please provide a detailed description. Additionally, after performing the Duncan's Multiple Range test for data analysis, it is necessary to indicate the differences between the treatments instead of just providing a p-value. Similar modifications are required for data in other tables. Furthermore, all instances of "p" in the manuscript should be italicized.

Response: We thank the reviewer for pointing these parts out.

In the Material and method Section, in the subsection “Essential oils”, we cleared this point. As Control 1 we used water, as Control 2 only acetone.

Line 347-348: “As a control, water (control 1) and acetone (control 2) were applied.”

The P value given in the tables refers to the one-factor analysis of variance, which was first calculated in order to determine if there was a statistically significant difference between the applied treatments and the two controls. After that, post hoc testing using Duncan's test was carried out to search for significant difference among treatments (concentrations). The results of the Duncan test are not shown in the table due to the scope of the analysis, but they are exposed in the text of the Results Section.

All instances of "p" in the manuscript have been italicized.

Comment 3:Table 2.

Additional details regarding the fumigant-contact and contact tests should be provided to clarify which gender exhibited higher toxicity based on the existing data.

Response: Thank you for your suggestion. We have revised the data you refer and made sums of dead flies considering different genders in a new table (Table 2) with the requested parameters. Based on the sums of dead female and dead males according to concentrations, it is easy to determine difference between controls and treatments or among treatments.

Line 111-113: “Considering gender, in the contact test females were slightly more resistant to treatments compared to males, since slightly less female flies dies compared to the males (Table 2).”

We observed the number of dead insects per replication, where 0 indicates no difference in dead male and female insect deaths, 1 indicates a higher number of dead male insects, and 2 indicates a higher number of dead female insects per replication. Then, a chi-square test was performed to determine whether there was a statistically significant difference in the number of dead flies between males and females. Table 2 shows the sums of dead females and males for all 4 replications, as well as chi-square tests. According to data shown in Tab. 2, it can be concluded that, for example, in the contact test with certain oils females were slightly more resistant to treatments compared with males (p<0.05).

Comment 4:Table 4.

More detailed data should be presented, including the number of hatched females and males.

Response: Thank you for your suggestion. We have revised the data and made sums of emerged flies considering different genders in a new table (Table 4) with the requested parameters.

Comment 5: Table 6.

It is necessary to calculate the total egg production per female under different treatments. The significant difference in total egg production among the control groups in various essential oil treatments needs to be explained. For instance, in the Geranium experiment, Control 2 had a total egg production of 29.25, whereas in the Dill experiment, Control 2 had only 20. An explanation for this variation should be provided.

Response: Thank you for pointing this out. We have included means of total egg production in Table 6.

The age and health of a female fly can affect its reproductive capacity. Since males and females were kept in the same cage, we cannot know if the used females in the experiment had already been mated. Genetic difference among individual fruit flies can result in a variation of their reproduction rate. Some flies may have genetic traits that make them more prolific egg layers, while others may lay fewer eggs. Environmental factors were always the same so it is likely that these factors were not involved.

We have explained it in Lines 174-175: "Egg production varies also on age, genetic difference, and health of a female fly, or if previous mating of adults had occurred.

Comment 6: Table 7.

Calculating the emergence rate under different treatments may be more scientifically sound, considering the varying egg production among females. A direct comparison of emerged adults may not be meaningful. The same modification should be applied to Table 3.

Response: Thank you for this suggestion. You have raised an important point here. Therefore, we have made a new table (Table 8) with emergence rate data. The calculation of the emergence rate in the contact toxicity test (Table 3) was not applicable because of difficulties raised in counting eggs in the medium, as we were not aware we had to dye the medium.

Comment 7:Discussion.

The current discussion section needs significant revision. For instance, in lines 384-398, despite extensive discussion, the intended conclusions of the authors are not clear.

The results of the GC-MS analysis are crucial supporting data and of great interest to readers. A detailed analysis of which components in the essential oils might play a primary role should be included.

Please incorporate these revisions to enhance the clarity and scientific rigor of the manuscript.

Response: Thank you for pointing this out. We agree with this comment. We have enriched Discussion Section especially in the part about the components of essential oils that might play a primary role, since this argument had not sufficiently been addressed. Also, we have updated discussion to show in a clearer way difference between tested oils.

Lines 246-249: The identified major compounds found in our study were citronellol (22.16%) and geraniol (21.22%), which are most likely responsible for the insecticidal activity against D. suzukii as well [36, 59].  

Lines: 272-276: Comparably, the compound isobornyl acetate may act as a repellent, but there is no available literature regarding P. silvestris EO effect on adults of D. suzukii, which suggests further testing of this essential oil as a deterrent and repellent against D. suzukii flies.

Comparing the four tested oils, it can be concluded that bergamot EO should be excluded from further testing. In the fumigant-contact and contact test with higher dose applied levels, all the other three oils were able to determine high level of mortality, from 92.5% to 100%. In the bioassays of repellent compounds, these oils showed a repellent effect after 24 hours.  The finding that dill and scots pine EOs can firstly be attractive to flies, can be considered interesting and needs to be studied further since it can be an aspect in attract and kill strategies. Geranium and dill EOs showed high deterrent effect even at the lowest applied doses (1% and 5%), with a mean of laid eggs less than one, compared with scots pine, which showed high level of detergency at high applied doses (10%). (repetition). When we compare the chemical composition of these 3 oils, we can affirm that their compositions are rather different, however, all three show insecticidal activity against D. suzukii.  A good strategy for developing and formulating bioinsecticide may be the combination of these oils they probably have different modes of action on this pest.

Standing the above considerations, EOs are a big group of biologically active substances with insecticidal effects and a wide arsenal of different modes of action, which is especially important for delaying D. suzukii development or avoiding its resistance to insecticides.

Reviewer 3 Report

In this MS, the authors carried out a series of experiments to study the insecticidal activity of selected four types of essential oils Pelargonium graveolens, Anethum graveolens, Pinus sylvestris and Citrus bergamia) against Drosophila through 4 different tests (contact toxicity, fumigant-contact toxicity, repellent effect and multiple-choice test). The topic is interesting. While there are some shortcomings which were following as:

  Q1: 2.5. Repellent effect: This is an unreasonable experimental design to set up the fixed placement of EOs on the left side and control on the right side. Random placement should be setup for the EOs and Control.

  Q2: 2.5: Repellent effect: “The position of flies is recorded (number of flies in test arm and number of flies in control arm) after a certain time of exposure” Here, how much the number of files in the central tube (which are no moving to test or control arm)? And because there are some files no moving and remaining in the central tube, the total number of number of flies in test arm + number of flies in control arm may be not same for the different treatments of Eos with three dese levels. So the preference index can not be used to compare with the significant differences among different treatments.

  Q3: Results: For 3.2. Contact toxicity, mortality of adult flies in fumigant-contact and contact bioassay, and 3.2.1. Emergence of the adults after exposure to contact toxicity were given. Why the emergence of the adults after exposure to fumigant-contact toxicity was not given?

  Q4: Results: 3.2. Contact toxicity: Give the subtitle 3.2.1 Mortality of adult flies in fumigant-contact and contact bioassay. And the primary 3.2.1 should be changed as 3.3.2!

  Q5: Table 5: No significant differences were given among different EOs at same dose level or among different dose levels for same EO. And while different active (A or R) at different dose level for same EO? What the meaning of this mechanism? Same question for the Table6.

  Q6: Table 9: No significant differences were given among different treatments of EOs.

Author Response

Reviewer 3:

Comment 1: In this MS, the authors carried out a series of experiments to study the insecticidal activity of selected four types of essential oils Pelargonium graveolens, Anethum graveolens, Pinus sylvestris and Citrus bergamia) against Drosophila through 4 different tests (contact toxicity, fumigant-contact toxicity, repellent effect and multiple-choice test). The topic is interesting. While there are some shortcomings which were following as:

Response:We thank the referee for the careful and insightful review of our manuscript.

Comment 2:Q1: 2.5. Repellent effect: This is an unreasonable experimental design to set up the fixed placement of EOs on the left side and control on the right side. Random placement should be setup for the EOs and Control.

Response: Thank you for this suggestion. You have raised an important point here. We considered that changing the position of control and the treatment arms would not have had an effect on the movement of insects or the preferendum index since the olfactometer was cleaned after each experiment and the length of the arms was the same. Thank you very much for pointing this out, it is very helpful. We will take it into account in future experiments.

Comment 3:Q2: 2.5: Repellent effect: “The position of flies is recorded (number of flies in test arm and number of flies in control arm) after a certain time of exposure” Here, how much the number of files in the central tube (which are no moving to test or control arm)? And because there are some files no moving and remaining in the central tube, the total number of number of flies in test arm + number of flies in control arm may be not same for the different treatments of Eos with three dese levels. So the preference index cannot be used to compare with the significant differences among different treatments.

Response:Thank you for pointing this out. Flies in the central tube are also calculated in the control, we have added it in the text. We have presented these results graphically in Figure S1, for an easier reading. We did not want to list a lot of tables in the manuscript, so we decided for the simpler method in this particular case with a repellent test (PI, symbols for attractive, neutral, and repellent activity, as well as graphical charts).

Line 393-395: The position of flies is recorded (number of flies in test arm, number of flies in control arm and in the central tube) after a certain time of exposure and a preference index is calculated. Flies in the central tube were calculated in the control.

Comment 4:  Q3: Results: For 3.2. Contact toxicity, mortality of adult flies in fumigant-contact and contact bioassay, and 3.2.1. Emergence of the adults after exposure to contact toxicity were given. Why the emergence of the adults after exposure to fumigant-contact toxicity was not given?

Response: Thank you for pointing this out. It is an interesting point to focus on in future studies, since this study has the intention to find essential oils that can be formulated as bio insecticides to be used on crops in fields. Therefore, we have only focused on monitoring the oviposition in the contact test since we consider this test as more suitable to the conditions of field. We thought that following oviposition in the fumigant-contact test did not have practical importance in this particular study done on adults.

Comment 5:Q4: Results: 3.2. Contact toxicity: Give the subtitle 3.2.1 Mortality of adult flies in fumigant-contact and contact bioassay. And the primary 3.2.1 should be changed as 3.3.2!

Response:We agree with this comment. Therefore, we have updated subsections and table names according to your suggestions.

2.1. Mortality of adult flies in fumigant-contact toxicity bioassay

2.2. Mortality of adult flies in contact toxicity bioassay

Table 1. Mortality of adult flies in fumigant-contact and contact bioassay.

Table 2. Gender differences in mortality in the fumigant-contact and contact bioassay

2.2.1. Emergence of the adults after exposure to contact toxicity bioassay

Table 3. Mean number of emerged adults after exposure to contact toxicity bioassay.

Comment 6: Q5: Table 5: No significant differences were given among different EOs at same dose level or among different dose levels for same EO. And while different active (A or R) at different dose level for same EO? What the meaning of this mechanism? Same question for the Table6.

Response: Thank you so much for this suggestion. The First part of the question is addressed in the response to Comment 3.

For bergamot EO, since the results are so different according to different concentrations and evaluation times, we think that this oil has no effect. The insects moved normally between control and treatments without any effect of the oil.

For the other EOs, which are mostly repellents, only at the beginning of the trial, some of them became attractive or neutral because it is possible that they were attracted by some scented component, but later on the repellent action became more determinant. This observation can be a useful tool for the development of attract and kill strategies.

Comment 7:  Q6: Table 9: No significant differences were given among different treatments of EOs.

Response: Table 9 refers to the chemical composition of the essential oils. (We believe that the reviewer thought on providing significant different for the table above).

Based on the results of descriptive statistics (mean and standard deviation), differences between the essential oils were commented in the Discussion Section which we enlarged to include your suggestion. Namely, on the basis of arithmetic means, it was clearly observed that geranium had the best effect at the lowest concentration. Due to the scope of the research, an analysis of variance was not performed to find differences in essential oils, but only for different concentrations of each essential oil, because additional such an analysis would have required 6 more tables and made the manuscript less readable.

The revised part where we discuss difference between EOs:

Comparing the four tested oils, it can be concluded that bergamot EO should be excluded from further testing. In the fumigant-contact and contact test with higher dose applied levels, all the other three oils were able to determine high level of mortality, from 92.5% to 100%. In the bioassays of repellent compounds, these oils showed a repellent effect after 24 hours.  The finding that dill and scots pine EOs can firstly be attractive to flies, can be considered interesting and needs to be studied further since it can be an aspect in attract and kill strategies. Geranium and dill EOs showed high deterrent effect even at the lowest applied doses (1% and 5%), with a mean of laid eggs less than one, compared with scots pine, which showed high level of detergency at high applied doses (10%) (repetition). When we compare the chemical composition of these 3 oils, we can affirm that their compositions are rather different, however, all three show insecticidal activity against D. suzukii. A good strategy for developing and formulating bioinsecticide may be the combination of these oils since they probably have different modes of action on this pest.

Standing the above considerations, EOs are a big group of biologically active substances with insecticidal effects and a wide arsenal of different modes of action, which is especially important for delaying D. suzukii development or avoiding its resistance to insecticides.

Round 2

Reviewer 2 Report

The authors have addressed all my concerns. The quality of the manuscript has been greatly improved.

Reviewer 3 Report

This version has been revised based on the comments of the reviewers.

Suggest to accept this MS.